# Centrosome Movements Are TUBG1-Dependent

**DOI:** 10.3390/ijms241713154

**Published:** 2023-08-24

**Authors:** Darina Malycheva, Maria Alvarado-Kristensson

**Affiliations:** Molecular Pathology, Department of Translational Medicine, Skåne University Hospital, Lund University, 21428 Malmö, Sweden; darina.malycheva@med.lu.se

**Keywords:** TUBG, TUBG meshwork, γ-tubule, γ-strings, centrosome

## Abstract

The centrosome of mammalian cells is in constant movement and its motion plays a part in cell differentiation and cell division. The purpose of this study was to establish the involvement of the TUBG meshwork in centrosomal motility. In live cells, we used a monomeric red-fluorescence-protein-tagged *centrin 2* gene and a green-fluorescence-protein-tagged *TUBG1* gene for labeling the centrosome and the TUBG1 meshwork, respectively. We found that centrosome movements occurred in cellular sites rich in GTPase TUBG1 and single-guide RNA mediated a reduction in the expression of TUBG1, altering the motility pattern of centrosomes. We propose that the TUBG1 meshwork enables the centrosomes to move by providing them with an interacting platform that mediates positional changes. These findings uncover a novel regulatory mechanism that controls the behavior of centrosomes.

## 1. Introduction

A centrosome is an independent membraneless organelle in animal cells [1]. It is composed of two microtubule-based barrel-shaped centrioles, which are surrounded by a network of proteins referred to as the pericentriolar matrix (PCM). In proliferating cells, the centrosome self-replicates simultaneously with the DNA and an interplay between centrosomes, and the mitotic spindle executes the segregation of offspring chromatids during mitosis [2]. In addition, the centrosome is a site that regulates the nucleation of actin and microtubules, and inherited centrioles can act as the basal body of flagella and cilia [1,2,3,4,5].

During interphase, the position of the centrosome changes in the cytoplasm [6,7], and its motion plays a role in cell differentiation and mitosis [6,8,9,10]. During DNA-replication, the cytosolic PCM shuttles between nuclear active origins of replication, which facilitates the transport of cytosolic proteins to the nuclear compartment [7]. In a cell-free assay, actin and microtubules were shown to limit the position of centrosomes [11]. In *Dictyostelium* and in primate epithelia kidney cells’ interphase, the central position of the centrosome is balanced with respect to forces generated by interactions with dynein, the radial microtubule array, and the cell cortex [12,13]. The basal bodies of multi-ciliated epithelia contain myosin II, which is needed for basal body migration [14]. During G2, myosin II and actin filaments supports the separation of the two centrosomes prior to segregation of the replicated genome [15]. The development of the brain in mice is affected by nuclear and centrosome movements, and dysfunctional centrosomes deplete neural stem cells causing microcephaly (reduce brain size) [16,17]. Budding yeast monitors nuclear and cytoplasmic division via the localization of the GTPase Tem1 to the spindle pole body. Once the spindle pole body migrates to the neck, Tem1 is brought into contact with Lte1, which in turn activates Tem1 to execute mitotic exit [18]. In mammalian cells, the completion of cell division occurs after one of the centrosomes transiently moves to the growing furrow, and the absence of centrosomes affects cytokinesis [8]. Hence, centrosome movements may integrate spatial and temporal cues in proliferating cells, but our knowledge is limited regarding the proteins affecting centrosome movements.

Centrosomes are rich in a member of the tubulin superfamily of GTPases, i.e., gamma-tubulin (TUBG) [19]. In mammals, TUBG1 is a ubiquitously expressed protein that is considered essential [20,21,22,23], whereas TUBG2 is expressed in the brain and during embryonic development [20]. TUBG occurs in the centrosome, cytosol, nucleus, and mitochondria [23,24,25,26]. The disruption of the centrosomes or decreased levels of TUBG lead to G1 arrest [7,25,27,28]. The polarity, dynamics, and positioning of the mitotic spindle depend on the presence of TUBG-containing microtubule-nucleating complexes, named the TUBG ring complex (γTURC), at the PCM [29]. In the present study, we assess the involvement of TUBG1 in the spatiotemporal localization of the centrosome. We find that centrosome movements occur in TUBG-rich areas, and reduced TUBG protein levels affect centrosome movements.

## 2. Results

### 2.1. The Centrosomes, γ-Strings, γ-Tubules, and the Nuclear Envelope Interact

The self-polymerizing capacity of TUBG creates a cellular meshwork consisting of TUBG threads and fibers called γ-strings and γ-tubules, respectively. Within the PCM, previous work has demonstrated that TUBG–pericentrin complexes form strings [30]. During nuclear assembly, γ-strings serve as a supporting scaffold for the formation of the nuclear compartment [26] and provide mitochondria with a cytoskeletal element that gives form to the mitochondrial network [23]. γTURCs, together with the centrosomal protein pericentrin, form γ-tubules [22], which have been shown to be most abundant in non-dividing cells (Figure 1) and have been suggested to act as a storage depot for cytosolic TUBG [22]. Confocal microscopy analysis of Z-stack images of whole fixed cells immunofluorescence-labeled with an anti-TUBG antibody show that endogenous γ-tubules, nucleus, and centrosomes are often in close vicinity to each other (Figure 1). γ-tubules can emanate from centrosomes (Figure 1a–c) or be positioned on the side (Figure 1a,b). These TUBG-rich components are often in direct contact with the nuclear envelope (Figure 1c) and can pass through the nucleus (Figure 1d), forming a spatio-temporal nuclear scaffold. These data confirm a direct interaction between TUBG-rich components and the TUBG-pool associated with the nuclear membrane [31,32].

### 2.2. Centrosome Moves in γ-String- and γ-Tubule-Rich Areas

A γ-tubule is a polar structure of variable length and mutations that affect the GTPase domain of TUBG impair its formation [22]. The analysis of γ-tubule formation has shown that γ-tubules nucleate on TUBG foci and on the nuclear envelope and have a low regrowth rate [22]. In contrast, γ-strings are non-polar structures formed from TUBG aggregates that are folded into γ-strings by the chaperone TCP-1 (CCT) [34].

To better understand the interactions among γ-tubules, centrosomes, and the nuclear envelope, we monitored live U2OS cells that stably expressed *TUBG* short hairpin RNA (shRNA; decreased the endogenous TUBG pool by ~40–50%) and co-expressed a C-tagged *TUBG1* green fluorescence protein (GFP) shRNA-resistant gene (which compensates the *TUBG* shRNA-mediated reduction of TUBG; Figure 2a) by recording Z-stack images in a time-lapse series for capturing complete U2OS-sh*TUBG*-GFP-TUBG cells. Fluorescence images show that the interphase centrosome can move along γ-tubules and often moves in close proximity to and around the nuclear envelope (Figure 2b and Appendix A) [35]. As γ-tubules, the PCM, and the nuclear envelope are TUBG-rich structures; this suggests that TUBG might facilitate the movements of a centrosome.

### 2.3. Visualizing Centrosomes in Live Cells

At the G1–S transition, a centrosome is composed of two centrioles and, adjacent to each centriole, a procentriole forms, which is the start of the formation of two centrosomes (Figure 3). At G2–M, the two centrosomes separate and the newly formed centrioles mature (Figure 3a).

To monitor the centrosome movements in live U2OS cells, we expressed the centriole marker centrin 2 gene N-tagged with a monomeric red fluorescence protein (RFP) in U2OS-sh*TUBG*-GFP-TUBG cells (Figure 3; U2OS-sh*TUBG*-GFP-TUBG-centrin 2) [8,36,37]. In living cells stably expressing the RFP-centrin 2 gene, we found that the resulting recombinant RFP-centrin 2 protein localized to the interphase centrosomes, in which two RFP-dots (centrioles) could be detected (Figure 3a). The monitoring of centrosome movements with a combination of time-lapse (every 150 s during 125 min) and Z-stack confocal microscopy further confirmed that the RFP-centrin 2-labeled centrioles followed the movements of the GFP-TUBG-labeled centrosomes during interphase, demonstrating that the location of RFP-centrin 2 can be used as a centrosomal marker in living cells (Figure 3b and Appendix A).

### 2.4. TUBG1 Knockdown Affects Centrosome Movements

Since γ-strings are often observed to emanate from a centrosome as it moves, and TUBG-rich elements are often in the vicinity of centrosomes (Figure 2 and Figure 3), we gene-edited the *TUBG1* gene using a single-guide (sg) RNA (green fluorescence protein (GFP)-tagged Cas9-CRISPR) targeting the *TUBG1* gene in U2OS cells (U2OS-sg*TUBG*) to determine whether TUBG affects centrosome motility. After four days of sg*TUBG* expression, the mean reduction in TUBG expression in U2OS-sg*TUBG* was ~40%, but this reduction was not homogeneous throughout the cell (Figure 4a). The immunofluorescence analysis of U2OS-sg*TUBG* cells shows a clear accumulation of TUBG in the nucleus (Figure 4a). Note that an sgRNA-induced reduction of TUBG is cytotoxic when TUBG protein levels drop below 50% [23]. Thereafter, we co-expressed the centriole marker RFP-centrin 2 and sg*TUBG* in U2OS cells (U2OS-sg*TUBG*-centrin 2; Figure 4b and Appendix A) [8,23,36]. After four days of co-expression, we monitored spatial and temporal changes in the position of centrosomes over 125 min using a combination of time-lapse (every 150 s) and Z-stack confocal microscopy. We then fixed the cells and subsequently immunostained them with an anti-TUBG antibody to confirm the colocalization of centrosomes with RFP-centrin 2-labeled centrioles. In the time-lapse images of living U2OS-sg*TUBG*-centrin 2 cells, the motility of the RFP-centrin 2-labeled centrioles was reduced in comparison to U2OS-sh*TUBG*-GFP-TUBG-centrin 2 cells (Figure 4c). The centrioles of U2OS-sg*TUBG*-centrin 2 cells are often located on the cytosolic side of the nuclear envelope. Their changes in position are modest and follow the movements of the nuclear envelope, without adopting a clear direction (Figure 4c). Overall, the data presented here demonstrate that reduced levels of TUBG impair the motility of centrioles, showing that the TUBG meshwork is necessary for the mobility of the centrosomes.

## 3. Discussion

Centrosomes are considered to be signaling and organizing centers that nucleate cytoskeletal elements such as microtubules, actin, vimentin, and γ-tubules [3,4,5,39]. Their motility is described to be mandatory for the performance of cellular duties [6,7,8,9,10,18,40]. Myosin II, dynein, microtubules, and actin are various proteins whose activities in previous studies have been connected to the positioning and motility of centrosomes [11,12,14,15].

TUBG is an important component of centrosomes that has been shown to be required for their function, structure, and replication [22,23,29,37,41,42]. Also, as shown in the 3D surface images presented in Figure 1d, there is a spatial relationship between the centrosomes, γ-strings, γ-tubules, and lamina structures, which provides a direct bridge at the nuclear envelope between the cytoplasm and the nucleus [7,26,31,32]. In addition, TUBG has also been reported to associate with the Golgi, the mitochondria, and the endoplasmic reticulum [23,43,44]. The presence of γ-strings in the membranes of mitochondria and the nucleus may provide a platform that could contribute to centrosome motility [23,26]. Accordingly, in live cells, centrosomes move close to the nuclear envelope and along γ-tubules, which are two TUBG-rich elements, and impaired expression of TUBG limits centrosome movements. A noteworthy finding is that the depletion of TUBG in cells is only partial, and the highest concentration of TUBG is found in the nucleus, next to which the centrosome lies. Altogether, these data are in accordance with the assumption that the TUBG meshwork works as a supporting platform.

There are many possible scenarios regarding how TUBG may facilitate the movements of a centrosome. One is the microtubule- and actin-nucleating capacity of TUBG as part of the γTURCs. The decreased protein levels of TUBG will most likely decrease the de novo formation of microtubules and actin fibers, which would also limit the microtubule- and actin-dependent functions of dynein and myosin II, respectively, resulting in a decrease in centrosome motility [11,12,14,15]. Similarly, possible motor proteins associated with γ-tubules may also promote changes in the positioning of centrosomes [22]. Another plausible scenario is the lateral interactions among the nuclear and the centrosomal γ-string boundaries and γ-tubules [26,30,32]. The formation of TUBG–TUBG bridges between structures may provide a sliding scaffold necessary for centrosome movements. Finally, γ-strings are non-polar structures, implying that γ-strings coming from the centrosome may connect the centrosome to the substratum and thus the concentration of TUBG facilitates and guides the direction of the movement. A finding supporting the latter statement is the lack of direction of the centrosomes upon impaired expression of TUBG1.

Nevertheless, the mechanisms and additional molecules that enable the TUBG-mediated motility of centrosomes require further analysis. Our findings demonstrate that the TUBG meshwork controls the changes in position of the centrosomes and identify TUBG as an important regulator of centrosome movements.

## 4. Materials and Methods

### 4.1. Reagents and cDNA

The following antibodies and reagents were used: anti-GFP (1:500, mouse sc-53882), Lamin B1 (1:1500); β-actin (1:5000, from Santa Cruz Biotechnology, Dallas, TX, USA); anti-TUBG (1:1000, rabbit T3320 and mouse T6557, from Sigma-Aldrich, St. Louis, MO, USA); donkey anti-mouse, rabbit, or goat conjugated to Cy3 (1:1600); Alexa488 (1:800) or Alexa647 (1:400) (Jackson ImmunoResearch, Cambridge, UK); and GFP-centrin 2, which was a gift from M. Bornens [8]. All other reagents were obtained from Sigma-Aldrich.

Human *TUBG* shRNA, pEGFP-sh-resistant *TUBG1*, pSG5-mRFP, and pSpCas9 (BB)-2A-GFP *TUBG1* sgRNA were prepared as previously described [22,25,37]. The *CETN2* gene 2 was amplified via PCR and subcloned into the EcoRI/BamHI sites of pSG5-mRFP using the following oligonucleotides: 5′GCGGAATTCATGGCCTCCAAC3′ and 3′GCGGGATCCTTAATAGAGGCTGG-5′.

### 4.2. Cell Culture, Transfection, and Western Blot Analysis

Human osteosarcoma U2OS cells and mammary gland epithelia MCF10A were cultured as previously reported [37,45]. The preparation of stably or transient transfected U2OS cells is described elsewhere [45]. The resulting cell lines were routinely tested for the presence of mycoplasma.

U2OS cells were examined on day 4 or for the indicated period of time after transfection with sgRNA [45,46].

Total lysates from cells were prepared and Western blot analyses were performed as described elsewhere [25,37,47].

### 4.3. Fluorescent Imaging Microscopy and Microtubule Regrowth Assay

The cell culture and fixation techniques used are described elsewhere [48]. All images were captured in a Zeiss LSM 700 Axio Observer microscope (ZEISS, Oberkochen, Germany) with a Plan-Apochromat × 63 NA 1.40 oil immersion objective. All images were subjected to rolling-ball background subtraction [49]. Time-lapse images were captured every 2.5 or 5 min for 125 or 300 min, and 10 sequential images were collected at 1 μm intervals.

A maximum-intensity projection of the sequential Z-stacks was analyzed with the TrackMate plugin [38]. In short, a sliding paraboloid method was applied to subtract the background of the image series before tracking. A DoG detector with auto-thresholding and a Simple LAP tracker were the settings used. The 3D viewer plugin (Fiji) was used to generate 3D surface images using default settings [33].

## Figures and Tables

**Figure 1 ijms-24-13154-f001:**
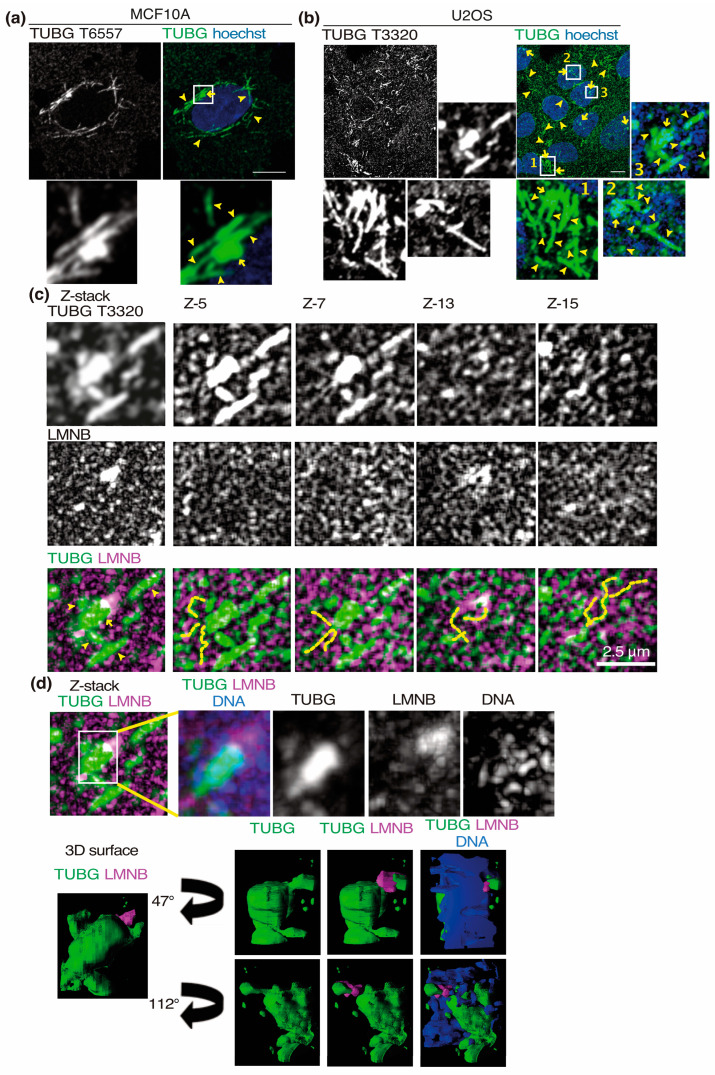
γ-Strings (dashed lines), γ-tubules (arrowheads), and centrosomes (arrows) are connected with the nuclear envelope. (**a**–**c**) MCF10A (**a**) and U2OS (**b**,**c**) cells were fixed, and endogenous TUBG was immunostained with an anti-TUBG antibody (mouse T6557; rabbit T3320); nuclei were detected with Hoechst, as indicated. Scale bars: 10 μm or as indicated. (**a**–**d**) Z-stacks show maximum-intensity projections of sequential images ((**a**) 30, (**b**–**d**) 21) that were collected at 0.2 μm intervals. The white box indicates the area magnified in the insets. (**c**,**d**) The sequential image series are from inset 3 in (**b**) and show chosen cropped Z-frames or maximum-intensity projections of sequential images of the interaction of γ-strings, γ-tubules, and centrosomes with the inner nuclear envelope marker lamin B (LMNB). (**d**) The cropped image stack (white box) of γ-strings, γ-tubules, and the centrosome were visualized in a 3D surface image with the 3D viewer plugin of ImageJ [33]. The 3D surface image obtained was rotated according to the indicated angle and the direction of the black arrow.

**Figure 2 ijms-24-13154-f002:**
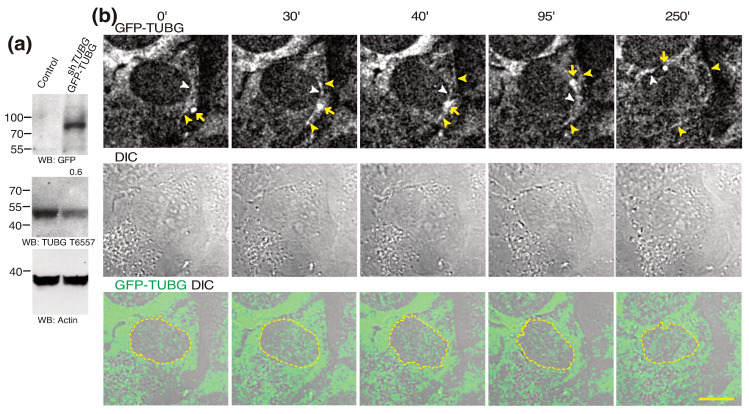
Centrosome moves in TUBG-rich areas in living cells. (**a**,**b**) U2OS and U2OS cells stably co-expressing *TUBG* shRNA (sh*TUBG*) and a *TUBG1* green fluorescence protein shRNA-resistant gene (GFP-TUBG; U2OS-sh*TUBG*-GFP-TUBG). (**a**) Total lysates from U2OS and U2OS-sh*TUBG*-GFP-TUBG were analyzed via Western blot (WB) with an anti-GFP (recognized GFP-TUBG), anti-TUBG (recognized endogenous TUBG), and anti-actin (loading control) antibodies, as indicated (*n* = 3). The number above the blot indicates the level of endogenous TUBG relative to control. (**b**) The differential interference contrast (DIC)/fluorescence images present one lapse of a time-lapse Z-stack sequential series of U2OS-sh*TUBG*-GFP-TUBG cells. Z-stacks show confocal images (maximum-intensity projection of 10 sequential images per time-point) that were collected at 1 μm intervals to enable the monitoring of the movements of the centrosome (yellow arrows) and its interactions with γ-tubules (yellow arrowheads) and the nuclear envelope (white arrowheads and dashed yellow borderlines). Scale bars: 10 μm. Related to Appendix A.

**Figure 3 ijms-24-13154-f003:**
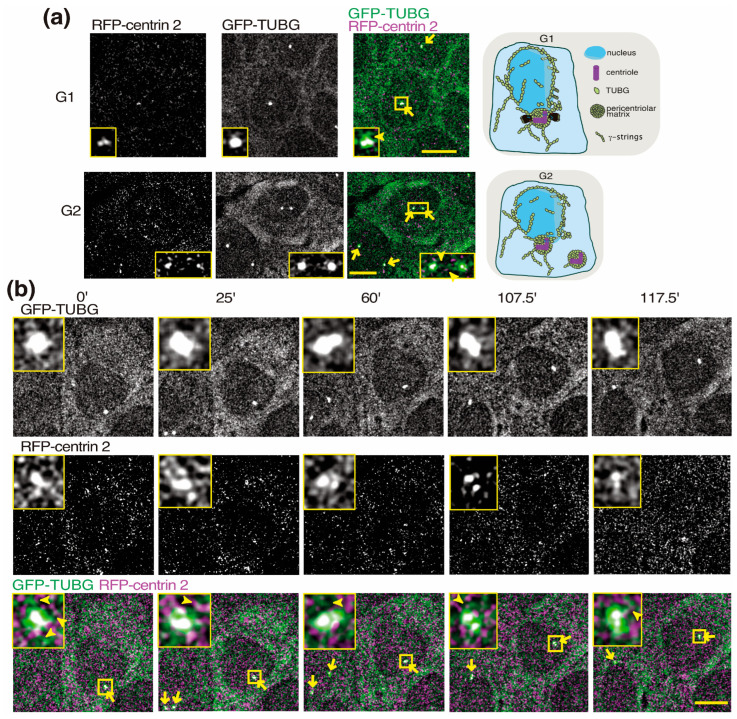
Monomeric red-fluorescence-protein-tagged (RFP)-centrin 2 labels centrosomes in live cells. (**a**,**b**) Each image presents one lapse of a Z-stack time-lapse series (maximum-intensity projection of 10 sequential images per time-point) of U2OS cells stably co-expressing *TUBG* shRNA and a *TUBG1* green fluorescence protein shRNA-resistant gene (GFP-TUBG) together with RFP-centrin 2 (to visualize the location of endogenous centrioles in a living cell; U2OS-sh*TUBG*-GFP-TUBG-RFP-centrin 2). Confocal images were collected at 1 μm intervals to determine the cellular position of centrosomes and centrioles (yellow arrows). The magnified areas in the insets (yellow boxes) show at least one centrosome containing two centrioles. Yellow arrowheads show γ-strings emanating from centrosomes. Scale bars: 10 μm. (**a**) Schematic representation depicting cells with two and four centrioles in G1- and G2-phase, respectively. (**b**) Each image presents one lapse of a Z-stack time-lapse series of U2OS-sh*TUBG*-GFP-TUBG-RFP-centrin 2. Related to Appendix A.

**Figure 4 ijms-24-13154-f004:**
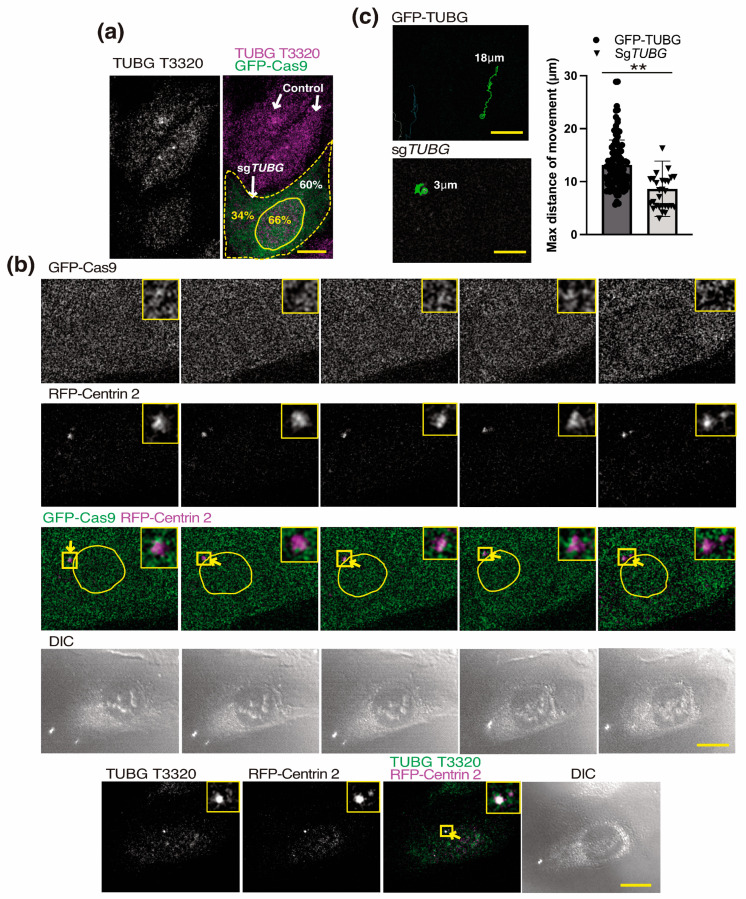
The cellular levels of TUBG1 regulate centrosome movements. (**a**) The confocal images are maximum-intensity Z-stack projections of 11 sequential image series that were collected at 0.6 μm intervals to determine the level of TUBG expression of whole fixed U2OS (control) or U2OS cells expressing a single-guide (sg) RNA (green-fluorescence-protein-tagged Cas9-CRISPR (GFP-Cas9); green, yellow dashed line) for four days targeting the *TUBG1* gene (sg*TUBG*; U2OS-sg*TUBG*). Samples were immunostained with an anti-TUBG antibody that originated in rabbit (TUBG T3320). Within the samples, quantification of TUBG was performed with ImageJ software by comparing immunofluorescently labeled TUBG1 in cells expressing GFP-Cas9 with non-expressing cells (control; *n* = 3). The white and yellow numbers in the image indicate the level of endogenous TUBG relative to control in the whole U2OS-sg*TUBG* cell, and the relative cytosolic and nuclear protein levels within the U2OS-sg*TUBG* cell, respectively. The yellow borderline marks the position of the nucleus. (**b**) The differential interference contrast (DIC)/fluorescence images present one lapse of a time-lapse Z-stack (maximum-intensity projection of 10 frames per time-point at 1.0 μm intervals) series of GFP-Cas9 expressing U2OS-sg*TUBG* cells that co-expressed RFP-Centrin 2 (U2OS-sg*TUBG*-Centrin 2; to visualize the location of endogenous centrioles (yellow arrows) in living U2OS-sg*TUBG* cells). The magnified areas in the insets (yellow boxes) show two RFP-centrin 2-labeled centrioles close to the nuclear envelope (yellow borderlines). Scale bars: 10 μm. Cell populations were viewed over a 125 min (50 frame) period before fixation and subsequent immunostaining with an anti-TUBG antibody to confirm the colocalization of RFP-centrin 2-labeled centrioles with the centrosome (yellow boxes). Related to Appendix A. (**c**) Images are the maximum-intensity Z-stack projection of the 50 recorded time-lapse images from the U2OS-sh*TUBG*-GFP-TUBG-centrin 2 (GFP-TUBG) and the U2OS-sg*TUBG*-Centrin 2 (sg*TUBG*) cells shown in Figure 3b and in b, respectively. Quantification of centriole movements was performed with ImageJ software via comparison of the changes in centriole position in U2OS-sh*TUBG*-GFP-TUBG-centrin 2 and U2OS-sg*TUBG*-Centrin 2 cells. The green lines are the recorded positions of the centrioles at 150 s intervals tracked with the TrackMate plugin [38]. The graph represents the recorded max distance that the centrioles have moved during the analyzed period (U2OS-sh*TUBG*-GFP-TUBG-centrin 2 *n* = 102 cells; U2OS-sg*TUBG*-Centrin 2, *n* = 29 cells; mean  ±  sd; ** *p*  <  0.01). Statistical analysis was performed using a two-way analysis of variance (ANOVA).

## Data Availability

Appendix A are publicly available at Zenodo (https://zenodo.org) under the accession number 8220349 (https://zenodo.org/record/8220349, accessed on 21 August 2023).

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
