# Peer review of "Centrosome Movements Are TUBG1-Dependent"

_ijms, 2023, doi:10.3390/ijms241713154_

Round 1

Reviewer 1 Report

In their manuscript entlited “Centrosome Movements are TUBG1 dependent” Darina Malycheva and Maria Alvarado-Kristensson aimed to establish the involvement of the TUBG meshwork in centrosomal motility. For this purpose the Authors labelled the centrosome by a monomeric red-fluorescence-protein-tagged centrin 2 gene and the TUBG1 meshwork by green-fluorescence-protein-tagged TUBG1 gene in living cells. Based on their data Malycheva and Alvarado-Kristensson suggest that the TUBG1 meshwork enables the centrosomes to move by providing them with an interacting platform that mediates positional changes. The topic of the paper is surely of interest since centrosome plays several very important roles in cell activity.

The main concern of this reviewer are the pictures.

Fig.1: I am not able to clearly identify the γ-Strings, γ-tubules , and centrosomes.  The Authors should use more explanatory images.

Fig.2b: the DIC picture is not clear. The Authors should insert a more contrasted image.

Fig. 3: in this figure this reviewer highlights a strong background. The Authors should clarify how to distinguish the signal of interest from the background.

Fig.4: also, in this case the DIC pictures should be more contrasted.

Another weakness concerns the Supplementary Materials because it was impossible to download the supporting informatiions at www.mdpi.com/xxx/s1. I'm very sorry about this because it would have been interesting to view the Authors’ videos.

Minor issues:

Introduction:

Pag 1 lines 20-21

I suggest changing this sentence because in this form it seems that centrosome acts as basal body. The Authors should clarify that the mother centriole may act as basal body to organize axoneme of cilia and flagella.

Pag 1 line 37:

“The development of the brain in mice is affected by nuclear and centrosome movements”.  This sentence seems inserted randomly in the text. Authors should better explain this very important point.

Results:

Pag. 4 line 76: Centrosomal not “centriosomal”.

Author Response

Reviewer #1 (Remarks to the Author):
In their manuscript entlited “Centrosome Movements are TUBG1 dependent” Darina Malycheva and Maria Alvarado-Kristensson aimed to establish the involvement of the TUBG meshwork in centrosomal motility. For this purpose the Authors labelled the centrosome by a monomeric red-fluorescence-proteintagged centrin 2 gene and the TUBG1 meshwork by green-fluorescence-protein-tagged TUBG1 gene in living cells. Based on their data Malycheva and Alvarado-Kristensson suggest that the TUBG1 meshwork enables the centrosomes to move by providing them with an interacting platform that mediates positional changes. The topic of the paper is surely of interest since centrosome plays several very important roles in cell activity.

The main concern of this reviewer are the pictures.

Fig.1: I am not able to clearly identify the γ-Strings, γ-tubules, and centrosomes. The Authors should use more explanatory images.

Author response #1

We have now labelled more structures in the presented images presented in figure 1.

Reviewer #1

Fig.2b: the DIC picture is not clear. The Authors should insert a more contrasted image.

Author response #2

As suggested, we have added more contrasted images.

Reviewer #1

Fig. 3: in this figure this reviewer highlights a strong background. The Authors should clarify how to

distinguish the signal of interest from the background.

Author response #3

In lines 168-175, it is explained that we defined as centrioles RFP-centrin 2 dots that followed the movements of the GFP-TUBG-labeled centrosomes. In lines 190-192, it is explained how we used immunostained endogenous TUBG of the fixed filmed cells for confirming the colocalization of RFP-centrin2 labelled centrioles with endogenous centrosomes. A disadvantage of using maximum-intensity Z-stack projections of sequential image series is that it might result in an increased background, but this method is necessary for keeping track of the centrosomes overtime in the whole cell.

Reviewer #1

Fig.4: also, in this case the DIC pictures should be more contrasted.

Author response #4

As suggested, we have added more contrasted images.

Reviewer #1

Another weakness concerns the Supplementary Materials because it was impossible to download the

supporting informatiions at www.mdpi.com/xxx/s1. I'm very sorry about this because it would have been interesting to view the Authors’ videos.

Author response #5

We are sorry to hear that it was not possible to download the uploaded files. We have uploaded new files. We hope that it will work this time.

 Reviewer #1

Minor issues:

Introduction:

Pag 1 lines 20-21

I suggest changing this sentence because in this form it seems that centrosome acts as basal body. The

Authors should clarify that the mother centriole may act as basal body to organize axoneme of cilia and flagella.

Author response #6

Thank you for your comment. As suggested, we have modified the sentence that is now placed in line 26.

Pag 1 line 37:

“The development of the brain in mice is affected by nuclear and centrosome movements”. This sentence seems inserted randomly in the text. Authors should better explain this very important point.

Author response #7

We have added additional text to the mentioned sentence (please see lines 38-39).

Results:

Pag. 4 line 76: Centrosomal not “centriosomal”.

Author response #8

Thank you for your helpful comment, we have amended the error.

Thank you for your helpful comments, time, and consideration.

Reviewer 2 Report

The manuscript of Malycheva and Alvarado-Kristensson entitled “Centrosome Movements are TUBG1 Dependent” deals with the role of the TUBG ring complex (γTURC) in centrosome movements. To this aim the AA used protein-tagged centrin 2 and protein-tagged TUBG1 genes for labeling the centrosome and the TUBG1 network. The AA suggest that the TUBG1 play an important role in centrosome movement, being involved in its spatiotemporal localization. The paper is potentially interesting since it try to decipher the mechanisms involved in centrosome positioning. However, I have some questions about the pictures that should be clarified.

Fig. 1a and b - How the AA can recognize the centrosome within the meshwork of gamma-tubulin unless a specific stain? The more concentration of gamma-tubulin within the meshwork is not sufficient.

Fig. 1c and d – How can the AA affirm that the arrow points the centrosome, whereas the arrowheads indicate only gamma-tubulin, but not centrosomal material? I think the Authors should stain the centrioles to definitively highlight the pericentriolar material.

Fig. 2b - γ-tubules and the nuclear envelope are not easily recognized.

Fig. 3a. Please explain why GFP-TUBG recognized here one or two distinct spots whereas TUBG T6557 stains fibers in Fig 1a.

Fig. 3b. I can see a diffuse spot-like staining for RFP-Centrin-2. I would have expected a distinct concentration on the centrosome/centrioles. In Fig. 4b the RFP-Centrin-2 labelling is clear and without background. Please explain.

l.205 – “… as shown in the 3D-surface images presented in Figure 1d, TUBG–TUBG interactions between the centrosomes, γ-strings, and γ-tubules enable the formation of TUBG–lamina structures..” These images may possibly show a spatial relationship between these structures, but it is difficult to argue that they could show a function.

l. 223 – “unknown motor proteins associated to γ-tubules may also promote changes in the positioning of centrosomes”. Unknown motor proteins: what kind of motor proteins? Different from dynein, kinesin, actin, myosin etc.? too vague concept, please explain.

Please indicate the secondary antibodies used and their concentration.

Please check the references: Abbreviations of the Journals; Authors names; etc.

I was unable to see the Supplementary Materials. I cannot open the files with the suggested address.

Author Response

Reviewer #2 (Remarks to the Author):

The manuscript of Malycheva and Alvarado-Kristensson entitled “Centrosome Movements are TUBG1 Dependent” deals with the role of the TUBG ring complex (γTURC) in centrosome movements. To this aim the AA used protein-tagged centrin 2 and protein-tagged TUBG1 genes for labeling the centrosome and the TUBG1 network. The AA suggest that the TUBG1 play an important role in centrosome movement, being involved in its spatiotemporal localization. The paper is potentially interesting since it try to decipher the mechanisms involved in centrosome positioning. However, I have some questions about the pictures that should be clarified. Fig. 1a and b - How the AA can recognize the centrosome within the meshwork of gamma-tubulin unless a specific stain? The more concentration of gamma-tubulin within the meshwork is not sufficient.

Fig. 1c and d – How can the AA affirm that the arrow points the centrosome, whereas the arrowheads

indicate only gamma-tubulin, but not centrosomal material? I think the Authors should stain the centrioles to definitively highlight the pericentriolar material.

Author response #1

I include the uncropped image presented in figure 1a. This is a maximum-intensity projection

of 30 sequential images. By capturing the sequential images, we can follow the changes in intensity in the y-axis and in this way, we can recognize the location of the centrosomes. As you see in the above image, the resulting Z-projection of all the images gives a very clear image of the meshwork, where the location of the centrosome can easily be established. There are no other structures in the size-range of the centrosomes that are stained with an anti-TUBG antibody.

Reviewer #2

Fig. 2b - γ-tubules and the nuclear envelope are not easily recognized.

Author response #2

It is easier if you have accessed to the video, where gamma-tubules and centrosome are labelled. We hope that you are now able to get access to the videos.

Reviewer #2

Fig. 3a. Please explain why GFP-TUBG recognized here one or two distinct spots whereas TUBG T6557 stains fibers in Fig 1a.

Author response #3

GFP-TUBG can be or not inserted in gamma-tubules. It is very difficult to record cells with GFP-labelled gamma-tubules. We think that, as the size of the resulting recombinant GFP-TUBG is bigger than TUBG, GFP-TUBG is less likely to be inserted in the fibers. In Figure 3a, there are not GFP-TUBG-labelled gamma-tubules. However, T6557 antibody can recognized both gamma-tubules as well as centrosomes. When we described the biology of gamma-tubules (Lindström, et.al., 2018; PMID: 29050966), we found that gamma-tubules are more abundant in cells in G1, that is, cells with one centrosome (one spot), and less abundant in G2, i.e., cells with two centrosomes (two spots).

Reviewer #2

Fig. 3b. I can see a diffuse spot-like staining for RFPCentrin-2. I would have expected a distinct concentration on the centrosome/centrioles. In Fig. 4b the RFP-Centrin-2 labelling is clear and without background. Please explain.

Author response #4

We have previously worked with GFP-centrin2 (PMID: 19648910). GFP-centrin2 is abundantly concentrated in the centrioles. Unfortunately, RFP-centrin2 is not abundantly inserted in centrioles. So, yes, we also hoped to obtain a better labelling of the centrioles. Regarding the background in the cells, there is not an easy answer to your question. Some cells have a higher expression level of RFP-centrin2 and thus have a higher background than cells with a lower expression level. We always try to capture cells with low expression levels of RFP-centrin2.

Reviewer #2

l.205 – “… as shown in the 3D-surface images presented in Figure 1d, TUBG–TUBG interactions between the centrosomes, γ-strings, and γ-tubules enable the formation of TUBG–lamina structures.”

These images may possibly show a spatial relationship between these structures, but it is difficult to argue that they could show a function.

Author response #5

Thank you for your comment, as suggested we have rewritten the sentence (please see lines 210 and 211).

Reviewer #2

  1. 223 – “unknown motor proteins associated to γ-tubules may also promote changes in the positioning of centrosomes”. Unknown motor proteins: what kind of motor proteins? Different from dynein, kinesin, actin, myosin etc.? too vague concept, please explain.

Author response #6

We have replaced unknown with possible (line 227). We wrote unknown as we do not know about possible motor proteins that might associate to gamma-tubules.

Reviewer #2

Please indicate the secondary antibodies used and their concentration.

Please check the references: Abbreviations of the Journals; Authors names; etc.

Author response #7

We have added the requested information (line 245).

Reviewer #2

I was unable to see the Supplementary Materials. I

cannot open the files with the suggested address.

Author response #8

We are sorry to hear that it was not possible to download the uploaded files. We have uploaded new files. We hope that it will work this time.

Thank you for your helpful comments, time, and consideration.

Round 2

Reviewer 2 Report

I think some images could be improved, but I think this aspect takes too much time

Author Response

Reviewer #2 (Remarks to the Author):
I think some images could be improved, but I think this aspect takes too much time

Author response #

We have once more looked att the DIC-images, to see if we could improve the images. In the current version of the manuscript, we have further increased the contrast of the DIC-images presented in Figure 2b.

Thank you for your comment, time and consideration.